# Motion Estimation in Coronary CT Angiography Images using Convolutional Neural Networks

Tanja Elss[#,⊗], Hannes Nickisch[#], Tobias Wissel[#], Rolf Bippus[#],
Michael M. Morlock[⊗], and Michael Grass[#]

[#]Philips Research, Hamburg, Germany
[⊗]Hamburg University of Technology, Germany

tanja.elss@philips.com

## Abstract

Coronary CT angiography has become a preferred technique for the detection and diagnosis of coronary artery disease, but image artifacts due to cardiac motion frequently interfere with evaluation. Several motion compensation approaches have been developed which deal with motion estimation based on 3-D/3-D registration of multiple heart phases. The scan range required for multi-phase reconstruction is a limitation in clinical practice. In this paper, the feasibility of single-phase, image-based motion estimation by convolutional neural networks (CNNs) is investigated. First, the required data for supervised learning is generated by a forward model which introduces simulated axial motion to artifact-free CT cases. Second, regression networks are trained to estimate underlying 2D motion vectors from axial coronary cross-sections. In a phantom study with computer-simulated vessels, CNNs predict the motion direction and the motion strength with average accuracies of $1.08°$ and $0.06$ mm, respectively. Motivated by these results, clinical performance is evaluated based on twelve prospectively ECG-triggered clinical cases and achieves average accuracies of $20.66°$ and $0.94$ mm. Transferability and generalization capabilities are demonstrated by motion estimation and subsequent compensation on six clinical cases with real cardiac motion artifacts.

## 1 Introduction

High quality CT imaging of the coronary arteries is a clinically important and challenging task. Hardware constraints restrict the temporal resolution of reconstructed CT image volumes and coronary motion artifacts frequently limit the diagnosis of coronary artery disease (CAD) or cause misinterpretations [1].

Motion vector field (MVF) estimation and subsequent motion compensated filtered back-projection (MC-FBP) [2; 3] are the key components of several motion compensation algorithms. Most of them deal with 3-D/3-D registration of multiple heart phases [4; 5]. This procedure requires an extended temporal scan range which corresponds to increased radiation doses. A motion estimation method for short scans has been introduced by Rohkohl et al. [6]. It is based on iterative minimization of motion artifact measures (MAMs) in a single reconstruced image volume. This iterative single-phase motion estimation method and the existing registration-based multi-phase approaches rely on raw projection data, respectively. This study addresses the question of how well motion estimation can be done from a single reconstructed CT image based on the coronary artifact shape.

1st Conference on Medical Imaging with Deep Learning (MIDL 2018), Amsterdam, The Netherlands.

Motion leads to differently shaped blurring artifacts depending on the angular reconstruction range and the motion trajectory during the acquisition (see Figure 1). A deep-learning approach is chosen for the image-based motion vector estimation, since feasibility of accurate motion artifact recognition and quantification at the coronary arteries by convolutional neural networks (CNNs) has already been shown in [7] and [8]. We utilize the proposed forward model from [8] for the introduction of simulated motion to artifact-free CT cases. On the basis of the resulting motion-perturbed image data, CNNs are trained for estimation of underlying motion vectors. Finally, the network behavior is analyzed to quantify the potential and limitations of single-phase, image-based motion estimation in clinical practice.

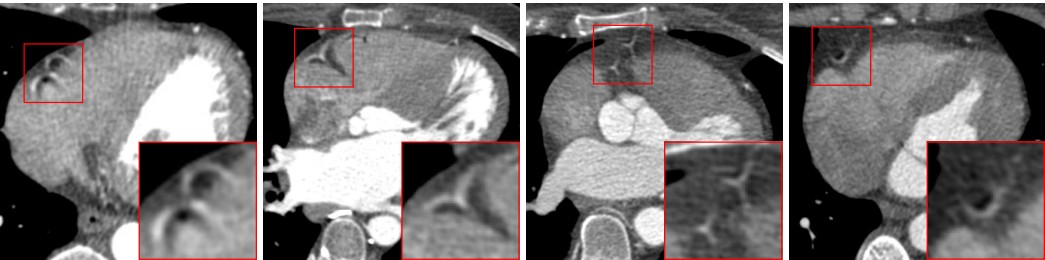

Figure 1: Examples of motion artifacts in the right coronary artery (RCA) are highlighted in four *step-and-shoot* cases. Arc-shaped blurring and intensity undershoots are typical artifact patterns.

## 2   Material

The forward models from [7; 8] require artifact-free CT cases as reference point determining the *no motion* state. As a first step, a phantom study is performed. Then, the transferability to clinical data is controlled based on twelve *step-and-shoot* cases. In the following, we detail the design of the computer-simulated vessels and the pre-processing of the clinical data.

**Phantom data**  A binary mask including three computer-simulated coronary arteries is created. Each vessel is modeled as a 3D cylinder whereby the centerline is oriented along the z-axis. The radii are set to 2, 3 and 4 voxels with an image resolution of 0.4 mm per voxel. Ray-driven forward projection [9] and subsequent high-pass filtering delivers the projection data required for application of the forward model. Projection geometry and corresponding ECG-data are adopted from clinical data.

**Clinical data**  Twelve prospectively ECG-triggered clinical data sets without coronary motion artifacts are collected by visual control. Acquisition was performed with a 256-slice CT scanner (Brilliance iCT, Philips Healthcare, Cleveland, OH, USA) and a gantry rotation speed of 0.272 sec per turn. The gating window for aperture-weighted cardiac reconstruction (AWCR) [10; 11] is chosen at mid-diastolic quiescent cardiac phase. The coronary centerline of each reconstructed CT image volume is segmented using the Comprehensive Cardiac Analysis Software (IntelliSpace Portal 9.0, Philips Healthcare, Cleveland, Oh, USA).

## 3   Method

CNNs are trained for motion estimation in axial coronary cross-sections. The required data for supervised learning is generated using an extended forward model for simulated motion introduction. Subsection 3.1 details the data generation process including motion vector field (MVF) creation and patch sampling. Data augmentation and data separation strategies as well as the supervised learning setups are described in Subsection 3.2.

### 3.1   Data generation

**Forward model**  The forward models from [7; 8] enable the generation of CT image data with controlled motion at the coronary arteries. They are based on the motion compensated filtered back-projection (MC-FBP) algorithm [3] taking artifact-free CT images and synthetic MVFs as input. In principle, arbitrary motion trajectories can be simulated by this approach by adjusting the synthetic MVF. For simplicity, we restrict the model to constant linear motion in the axial plane.

Therefore, minor adaptations of the simulated MVF with constant linear motion from [8] are performed. In the forward model, the displacement $\vec{d}_{\vec{c}}$: $[0\%, 100\%] \times \Omega \to \mathbb{R}^3$ of each voxel $\vec{\nu} \in \Omega \subset \mathbb{R}^3$ at time point $t_{cc} \in [0\%, 100\%]$ in millimeters is calculated by:

$$\vec{d}_{\vec{c}}(t_{cc}, \vec{\nu}) = s \cdot m_{\vec{c}}(\vec{\nu}) \cdot \vec{\delta}_{\vec{c}}(t_{cc}, \alpha) \tag{1}$$

As described in [8], $t_{cc}$ is measured in percent cardiac circle, $\Omega$ denotes the field of view, $m_{\vec{c}}$: $\Omega \to [0,1]$ is a weighting mask which limits the motion to the area of the currently processed centerline point $\vec{c} \in \Omega$ and $\vec{\delta}_{\vec{c}}$ determines the motion direction. For our purposes, $\vec{\delta}_{\vec{c}}$: $[0\%, 100\%] \times (-180°, 180°]$ is adapted to:

$$\vec{\delta}_{\vec{c}}(t_{cc}, \alpha) = \frac{60\text{bpm}}{\text{HR}_{\text{mean}}} \cdot \frac{\vec{\rho}_{\vec{c}}(\alpha)}{\|\vec{\rho}_{\vec{c}}(\alpha)\|_2} \cdot \begin{cases} -0.5 & \text{if } t_{cc} < r - 10\% \\ \frac{(t_{cc}-r)}{20\%} & \text{if } r - 10\% \leq t_{cc} \leq r + 10\% \\ +0.5 & \text{if } t_{cc} > r + 10\% \end{cases} \tag{2}$$

The parameter $\text{HR}_{\text{mean}}$ denotes the patients mean heart rate during acquisition and $r$ is the reference cardiac phase during AWCR. The motion direction determined by $\vec{\rho}_{\vec{c}}(\alpha)$ is limited to the axial plane, i.e. the z-coordinate is set to zero. The x-coordinate and the y-coordinate are chosen so that $\alpha$ corresponds to the angle between mean reconstruction direction of the currently processed centerline point $\vec{c}$ and motion direction. The mean reconstruction direction is defined by the gantry rotation angle at the reference heart phase $r$ and is constant for each voxel reconstructed by the same circular scanning shoot. It has to be noted, that the system rotation direction is equal for all cases. This is important, since the reverse rotational directions would lead to a flipping of the artifact shapes.

The papers [7; 8] investigate the feasibility of motion artifact recognition and quantification by utilizing the parameter $s$ for target value assignment. Compared to these works, our forward model has an additional (angular) degree of freedom $\alpha$, i.e. each MVF is now defined by a parameter tuple $(s, \alpha)$. The so-called target motion strength $s \in \mathbb{R}_+$ scales the length of each displacement vector in the MVF and therefore determines the motion width. On the basis of the velocity measurements at the coronary arteries by Vembar et al. [12], the target motion strength $s$ is limited to the interval $[0, 10]$ in the following experiments. The newly introduced angle parameter $\alpha \in (-180°, 180°]$ determines the in-plane motion direction. Both parameters $s$ and $\alpha$ are randomly sampled from uniform distributions in the following experiments. The corresponding Cartesian coordinates $x = s\cos(\alpha)$ and $y = s\sin(\alpha)$ are defined as ground-truth labels for the supervised learning task.

The extended forward model enables the generation of multiple motion-perturbed CT image volumes with controlled motion level and motion direction at a specific coronary centerline point $\vec{c}$. For each centerline point $\vec{c}$ and parameter setting $(s, \alpha)$, one 2D image patch is sampled as input data for supervised learning.

**Patch sampling:** An image patch of size $80 \times 80$ pixels is sampled from the axial plane with an image resolution of $0.4 \times 0.4$ mm$^2$ per pixel. The centerline point $\vec{c}$ defines the patch center and the patch is spanned by two orthogonal vectors which are constructed with respect to the mean reconstruction direction of the centerline point. By this procedure, the information about the angular reconstruction range is embedded in the patch orientation.

In the phantom study, the forward model with subsequent patch sampling is applied 2000 times per simulated vessel, thus, delivering a total amount of 6 000 samples as data base. Figure 2 shows coronary cross-sectional patches for varying parameter settings $(s, \alpha)$. Depending on the motion angle $\alpha$, differently shaped blurring artifacts occur. Orthogonal motion ($\alpha = \pm 90°$) leads to banana-shaped artifacts while parallel motion ($\alpha = 0°$ or $\alpha = 180°$) causes bird-shaped blurring.

In the clinical study, the forward model with subsequent patch sampling is applied 2000 times per step-and-shoot case. By this procedure, a total amount of 24 000 samples is collected as data base for supervised learning. It has to be noted that merely centerline segments with a maximal inclination of 45 degree to the z-axis are included to assure cross-section characteristics. The gray values are clipped to the relevant intensity range with a window/level setting of 900/200 HU and normalized to the interval [-1,1]. Figure 3 shows an example patch for varying parameter settings $(s, \alpha)$. Compared to the corresponding phantom semicircle plot in Figure 2 artifact shapes are visually more difficult to distinguish. Especially in case $\alpha = 0°$, increasing motion levels are hard to recognize. Visibility of blurring artifacts and intensity undershoots are strongly influenced by surrounding background intensities.

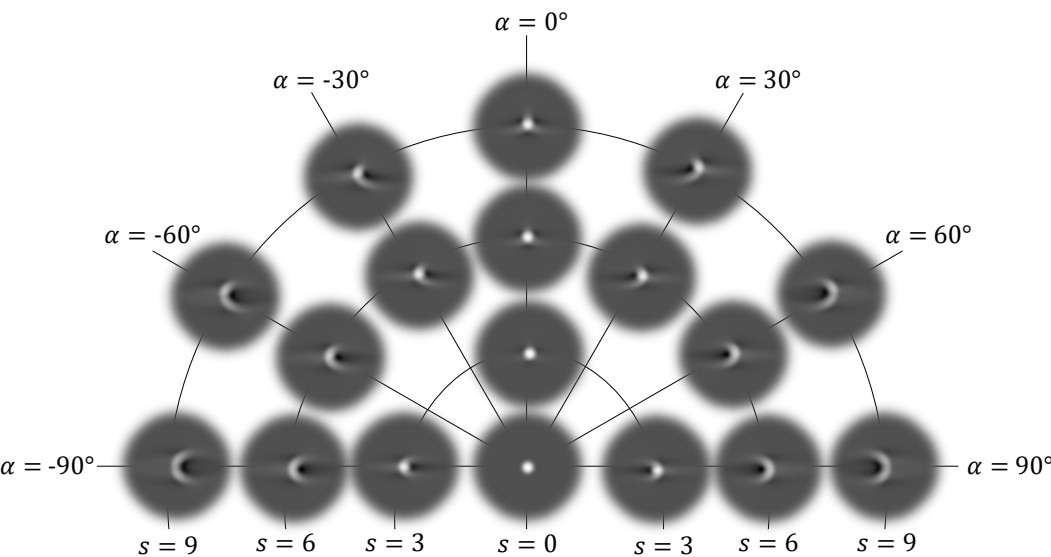

Figure 2: Computer-simulated vessels are motion-perturbed using the forward model with different parameter settings $(s, \alpha)$. For each setting a coronary cross-sectional image patch is sampled as input data for supervised learning. The angle parameter $\alpha$ determines the artifact shape while the artifact size is controlled by the motion strength parameter $s$. In general, most severe motion artifacts occur in case of orthogonal motion ($\alpha = \pm 90°$).

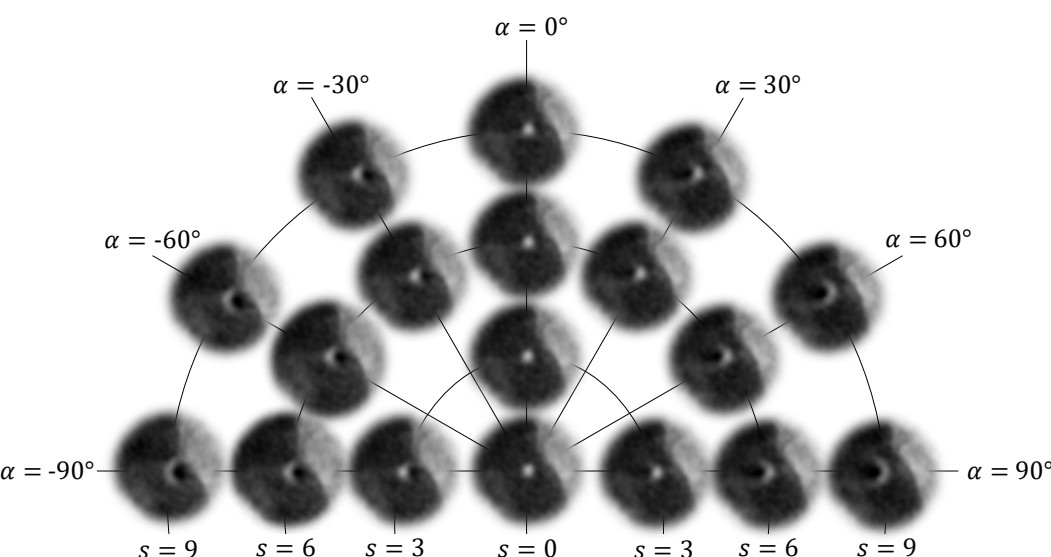

Figure 3: A clinical CT case is locally motion-perturbed at one coronary centerline point by the forward model with different parameter settings $(s, \alpha)$. The sampled coronary cross-sectional image patches show typical artifact pattern like arc-shaped blurring and intensity undershoots. In contrast to the phantom study, varying background intensities lead to lower visibility of the blurring artifacts.

## 3.2 Supervised Learning

**Data separation** Figure 4 illustrates two setups of separating the phantom data in training and validation subsets. In the angle partitioning, all samples with $\alpha \in \bigcup_{i=-2}^{1}[35° + i \cdot 90°, 55° + i \cdot 90°]$ are assigned for validation. The complementary set is used for training, i.e. angle interpolation capabilities are tested by this setting. In the same manner, all samples with $5 \leq s \leq 7$ are assigned for validation during strength partitioning.

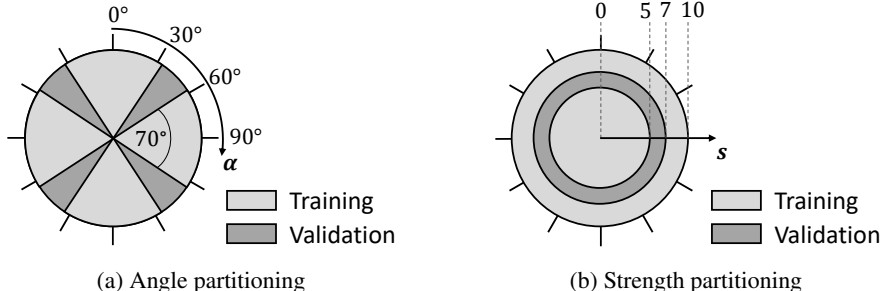

(a) Angle partitioning    (b) Strength partitioning

Figure 4: The phantom data is separated in training and validation subsets with respect to the parameters $\alpha$ and $s$. In both experimental setups interpolation capabilities are tested.

Table 1: Quantitative comparison of the validation results in the phantom and the clinical study.

| Data | $(x, y)$ error: $\varepsilon_{x,y}$ | $\alpha$ error: $\varepsilon_\alpha$ | $s$ error: $\varepsilon_s$ |
|---|---|---|---|
| Phantom (angle partitioning) | $0.088 \pm 0.078$ | $1.084° \pm 3.861°$ | $0.062 \pm 0.062$ |
| Phantom (strength partitioning) | $0.086 \pm 0.051$ | $0.559° \pm 0.450°$ | $0.053 \pm 0.042$ |
| Clinical | $1.497 \pm 1.200$ | $20.659° \pm 30.985°$ | $0.942 \pm 0.924$ |

Due to the patch similarity of adjacent centerline points, the clinical data is case-wise separated for training and validation with a ratio of $10 : 2$. In this way, robustness of the trained networks is evaluated with regard to unknown variations in the background intensities.

**Data augmentation** The data basis during network training is extended by online data augmentation. Symmetry properties are exploited for horizontal and vertical patch mirroring. The target labels $x$, $y$ are adapted accordingly. This procedure quadruples the amount of background variations in the clinical study. Additionally, cropping is performed as label-preserving augmentation strategy. Sub-patches of size $60 \times 60$ pixels are randomly selected in order to build translation invariance into the networks. This allows for slight variations in the in-plane coronary positions. During validation, the center patch is cropped and no mirroring is performed.

**Learning setup** The Microsoft Cognitive Toolkit (CNTK v2.0+, Microsoft Research, Redmond, WA, USA) is used as deep learning framework. A 20-layer ResNet [13] is selected as network architecture. The number of filters is doubled in each layer yielding $\{32, 64, 128\}$. The kernel size of the average pooling is increased to 15, according to the input data size. The last layer has a linear activation function and two output neurons to predict $\hat{x}$ and $\hat{y}$. The stochastic gradient descent solver Adam [14] with an initial learning rate of 0.05, a mini-batch size of 32 and a momentum of 0.8 is used for network optimization. The learning process is driven by the squared error $l = (x - \hat{x})^2 + (y - \hat{y})^2$. Training is performed over 60 epochs while the learning rate decreases with a factor of five after every 20th epoch. In the phantom study no regularization is performed. L2 regularization with a weight of 0.001 is used in the clinical study.

## 4 Experiments and Results

The following error metrics are introduced for network evaluation:

$$\varepsilon_{x,y} = \sqrt{(x - \hat{x})^2 + (y - \hat{y})^2}, \qquad \varepsilon_\alpha = \min(|\alpha - \hat{\alpha}|, 360° - |\alpha - \hat{\alpha}|), \qquad \varepsilon_s = |s - \hat{s}|$$

The validation results including mean and standard deviation of each error metric are summarized in Table 1. As expected, more accurate motion vector prediction is achieved during the phantom study. Several experiments are performed to improve performance on the clinical data. Separation of the tasks angle prediction and strength prediction by two independent network trainings does not lead to improved accuracy. Also, a network initialization with the learned weights from the phantom studies and subsequent fine-tuning on the clinical data provides no advantage over network optimization from scratch. The following paragraph deals with an error analysis of the network from Table 1 which is merely trained on clinical data.

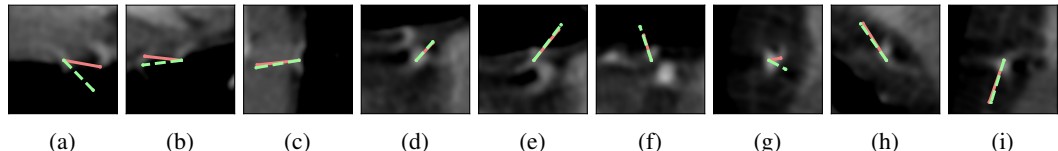

(a)      (b)      (c)      (d)      (e)      (f)      (g)      (h)      (i)

Figure 5: The predicted motion vectors of nine exemplary cross-sectional patches are visualized as red lines. The corresponding ground truth motion vectors (dashed line) are highlighted in green.

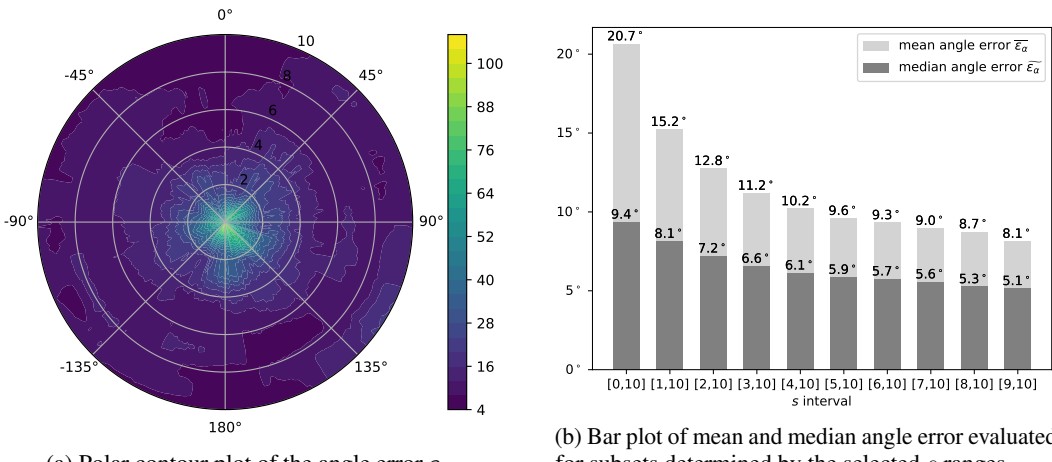

(a) Polar contour plot of the angle error $\varepsilon_\alpha$.

(b) Bar plot of mean and median angle error evaluated for subsets determined by the selected $s$ ranges.

Figure 6: The angle error $\varepsilon_\alpha$ is calculated for each image patch in the clinical validation data and visualized with regard to the $(s, \alpha)$ coordinate. High angle errors $\varepsilon_\alpha$ correlate with low $s$ values, i.e. most accurate prediction of the motion direction is feasible for image patches with severe motion artifacts.

**Error analysis** A qualitative error analysis is performed by visual inspection of the validation data. Example patches with corresponding predicted and ground truth motion vectors are visualized in Figure 5. Non-visible coronary blurring at the heart wall due to the performed intensity clipping is identified as one possible source of errors (see Figure 5a-5c) whereas overlapping artifacts at vessel bifurcations do not seem to confuse the neural network (see Figure 5d, 5e). As illustrated in Figure 5f, the CNN frequently delivers too conservative predictions of the motion strength $s$. In case of small or mid-size artifacts the predicted motion direction is less accurate (see Figure 5g). In contrast, the angle $\alpha$ is predicted with high accuracy in the presence of severe artifacts (see Figure 5h, 5i). A quantitative analysis of this observation is performed.

The error plots in Figure 6 illustrate the correlation between the accuracy of the predicted motion direction and the introduced motion strength $s$. Mean angle errors $\overline{\varepsilon_\alpha}$ of less than $10°$ are obtained in the clinical validation data for image patches with $s > 5$. Visual inspection disconfirms the need of motion compensation in case of $s \in [0, 1]$. Hence, initial tests for motion estimation and motion compensation on clinical data with real artifacts are performed.

**Motion compensation experiment** Six test cases with prospectively-gated acquisition mode and cardiac motion artifacts at the RCA are included. Based on the proposed deep-learning-based motion estimation, a motion compensation pipeline is developed. The key processing steps and the results of the motion compensation experiment are illustrated in Figure 7. In the first step, coronary cross-sectional patches are sampled as CNN input data with respect to the mean reconstruction direction according to Subsection 3.1. The approximate location of the coronary artery is selected manually. For each motion-degraded input patch, one 2D axial motion vector (highlighted in red) is predicted. The corresponding continuous MVF is calculated according to Equation (1). In our case of linear motion, the reversing motion trajectory $\vec{d}_{\vec{c}}^{\,-1}$ is obtained by multiplication with minus one. The resulting inverse MVF is included in the MC-FBP algorithm. The reconstructed image volume is locally motion compensated at a limited area around the manually selected center point.

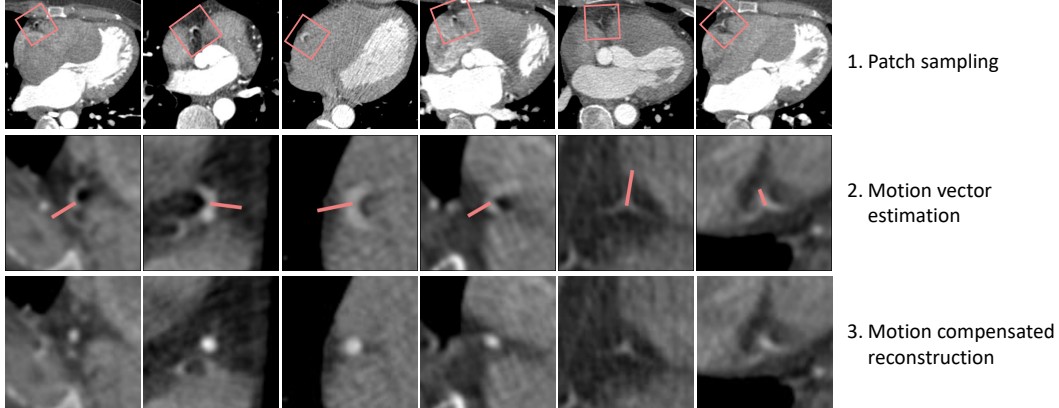

Figure 7: Coronary cross-sectional patches are sampled as CNN input data with respect to the mean reconstruction direction. MVFs are created by means of the predicted axial motion vectors. Subsequent motion compensated reconstruction by MC-FBP leads to reduced artifact levels in most of the cases.

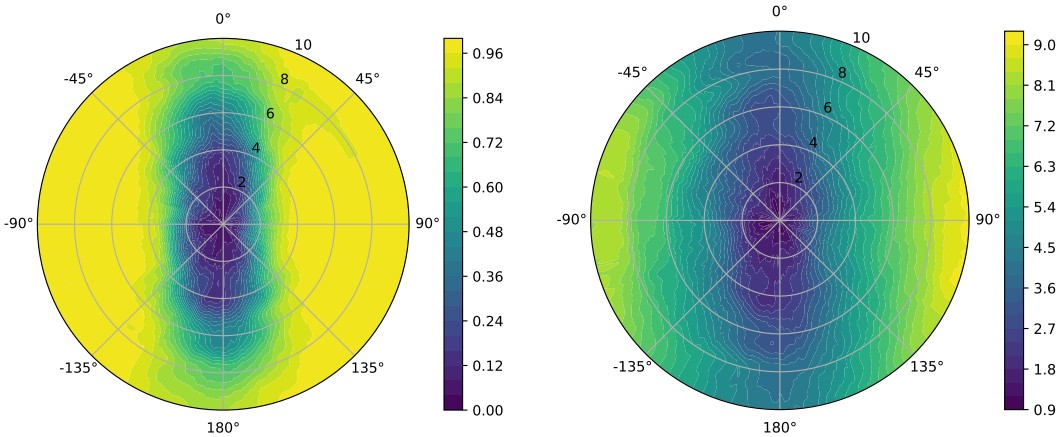

(a) Polar contour plot of the classification output.     (b) Polar contour plot of the regression output.

Figure 8: The predicted artifact probability of the classification network and the predicted artifact level of the regression network from [8] are calculated for the clinical validation data and visualized with regard to the underlying $(s, \alpha)$ coordinate. The networks diagnose least severe artifacts in case of parallel motion ($\alpha = 0°$ or $\alpha = 180°$). Especially, the output of the regression network is spatially smooth.

Motion artifact reduction is achieved in four out of six cases. Reduced arc-shaped blurring artifacts and intensity undershoots can be observed in case one to four. So far, the proposed motion compensation approach is focused on vessels near the patch center. Therefore, minor motion artifact introduction can be observed in neighboring anatomy. The bird-shaped blurring artifacts from case five and six are not removed after motion compensated reconstruction.

It has to be noted that the predicted axial motion vectors in cases five and six appear reasonable for a human observer. The missing improvement might be caused by the limitation to 2D axial motion compensation. In fact, more complex trajectories than constant linear motion are possible and motion in z-direction should also be considered. This assumption is encouraged by the artifact level analysis in Figure 8. As already mentioned in Section 3.1 bird-shaped blurring artifacts may be caused by parallel motion ($\alpha = 0°$ or $\alpha = 180°$), but increasing motion levels do not result in severe artifacts as in the case of orthogonal motion ($\alpha = \pm 90°$). This visual impression agrees with measured artifact probabilities (see Figure 8a) and artifact levels (see Figure 8b) of clinical validation patches by the CNNs from [8]. The present bird-shaped artifacts could merely be caused by unusually large axial

displacements. More likely is the appearance of turning motion during acquisition which also leads to bird-shaped artifacts.

Nevertheless, artifact reduction is achieved in most of the test cases, despite limitation to axial motion and simple linear compensation. This motion compensation experiment demonstrates generalization capabilities of the trained CNN, since transferability from simulated to real cardiac motion artifacts is shown.

## 5   Discussion

We proposed the first single-phase motion estimation approach which works solely on reconstructed image data. The carefully designed motion model which comprises linear trajectories in the axial plane, reveals potential and limitations of image-based motion estimation. Due to variations in noise level, background intensity, vessel structure and contrast agent density, accurate prediction is substantially more difficult in clinical cases compared to phantom data. The trained CNNs are remarkably successful in solving this ill-posed problem.

The proposed motion compensation experiment demonstrates that the trained network also achieves reasonable results on data with real (non-simulated) motion artifacts. This can be considered as proof of principle. Since solely one back projection step is required for explicit motion vector prediction, our method offers an advantage with respect to computational effort for image reconstruction compared to existing approaches. However, total running times and performances are not comparable so far. A lot of future research is required for establishment of a full 3D motion compensation method. This comprises model extension to more complex motion, i.e. including motion along the z-axis and turning motion. The introduced procedure of data generation by a forward model and subsequent supervised learning is, in principle, extendable to arbitrary non-linear 3D motion trajectories. It needs to be reviewed to which extent the information content of the reconstructed image volumes is a limiting factor for estimation of more complex trajectories. The benefit of additional information provided by 3D input patches as well as transferability to other scanner types and imaging protocols should be investigated.

The trained networks might also be integrated in existing motion compensation pipelines, e.g. by defining initial MVFs. A cascade process is conceivable. In a first step, the artifact measures from [8] could decide whether and where motion correction is required. For remaining patches with mid-size or severe artifacts, more accurate prediction of the motion direction can be performed. Robustness might additionally be improved by motion vector smoothing of adjacent centerline points in order to compensate for scatter and outliers.

## 6   Conclusions

Typical coronary artifact patterns are introduced in phantom and clinical data by a forward model which simulates linear, axial motion. The generated image data is used for subsequent supervised learning of CNNs for estimation of underlying motion vectors. The synthetically motion-perturbed data allows one to investigate the relationship between motion direction, angular reconstruction range and resulting artifact shapes. Most accurate prediction of the underlying motion direction is feasible for cross-sectional image patches with severe artifacts. A motion compensation strategy is developed to verify generalization capabilities of the trained CNN to motion estimation in clinical practice.

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
