# OpenReview forum: "Motion Estimation in Coronary CT Angiography Images using Convolutional Neural Networks"
_MIDL.amsterdam/2018/Conference — MIDL 2018 Poster_

### Review · AnonReviewer2 · 2018-05-09
**A very interesting and novel approach to correcting the coronary artery motion artifacts in CT. However, the validation of the method is very limited. The method was not empirically compared with any alternatives and theorethical advantages of it over alternatives are unclear to me.**

**Rating:** 3
**Confidence:** 1

**Review:**

First of all, I find it important to note that I know little about motion artifacts in CT and about methods for their correction, apart from what I could learn from this paper and a very shallow examination of some of the references.

Strengths:
+ The method presented in the paper is novel and very different from the other methods for solving the same problem (as far as I can see from the literature review in the paper). It has an advantage of needing only one CT image compared to some of the other proposed methods.
+ The paper is written rather well and is interesting to read.
+ The proposed method works well in the majority of the cases.
+ Various aspects of the algorithm's performance were studied (the relationship between angle and strength of motion and network's error when estimating those parameters; artifact probability and severity predicted by a network and the angle and strength).
+ I clearly see that a lot of work has gone into this research project.

Weaknesses:
- The validation of the method is rather weak. Motion correction performance was evaluated only in 6 scans. and only qualitatively. The motion estimation performance in clinical data with simulated artifacts is validated only using 2 scans (as far as I understood, no cross-validation was performed). In the phantom data experiment (both images and motion is simulated), as far as I understood, only the case when the artery is perpendicular to z was considered. In all cases, only 2D motion was simulated. To sum up: 1) the algorithm was validation on only 6 real cases, 2) the validation on artificial data is not very extensive.
- The advantages of the method over a method that also only needs short scan range [6] are discussed but do not sound convincing to me. As I understood, those are: 1) [6] relies on raw projection data, 2) [6] is more computationally expensive. (Written in the introduction and discussion). I do not understand why relying on raw projection data is bad, since it should be always available. (The discussion says: "Furthermore, the dissociation from the raw projection data opens up additional applications beside motion compensation. For instance, utilization for motion model establishment should be considered. In general, reconstructed image data is less limited in scope than raw projection data." I do not understand what this means.) Regarding the second stated advantage: CNNs are also computationally expensive and require GPUs to train and apply them. If the authors wish to make an argument of computational efficiency, they should compare the running times, at least theoretical ones.
- No performance comparison with a similar method - e.g. [6]. (Though, I am not entirely sure they are comparable, and if they are not it should be explicitly stated in the paper.)
- What is the difference between the paper and [7-8]? I am going to guess that in [7-8] are only about artifact recognition, whereas this paper is about MVF estimation. I think it should be stated more clearly.
- The paper is rather involved and it might be possible to make it more understandable, especially for people who are new to the problem. Here are things that I am not sure I understand or I didn't understand:
1) When I hear "scan range" I think about spatial range (e.g. which body part is examined?). I think in the paper it mostly means duration of the scan. I think it should be clarified.
2) What is multi-phase and single-phase reconstruction?
3) What is "reconstruction direction"?

**Special Issue:**

No

---

> ### Comment · ~Tanja_Elss1 · 2018-06-11
> **Re: A very interesting and novel approach to correcting the coronary artery motion artifacts in CT. However, the validation of the method is very limited. The method was not empirically compared with any alternatives and theorethical advantages of it over alternatives are unclear to me.**
>
> Thank you for your detailed comments and your thorough work. We revised our paper according to your remarks and we would like to address the mentioned weaknesses:
>
> 1.	Motion correction performance was evaluated only in 6 scans and only qualitatively.
> […] No performance comparison with a similar method - e.g. [6]. (Though, I am not entirely sure they are comparable, and if they are not it should be explicitly stated in the paper.)
>
> The proposed motion compensation experiment is solely a proof of concept and demonstrates that the trained network also achieves reasonable results on data with real (non-simulated) motion. Total running times and performances are not comparable so far. First, further research has to be done to establish a full 3D motion compensation method. Extension of our model to motion along the z-axis and turning motion will be the next steps. We stated this in the revised paper.
>
> 2.	The motion estimation performance in clinical data with simulated artifacts is validated only using 2 scans.
>
> We agree. Unfortunately, our data base is really limited in scope. It is hard to get clinical cases which meet our claims (without motion artifacts and with corresponding raw projection data). But it has also to be considered that each clinical case delivers a multitude of coronary cross-sections (2 cases correspond to 4000 validation patches).
>
> 3.	I do not understand why relying on raw projection data is bad, since it should be always available. (The discussion says: "Furthermore, the dissociation from the raw projection data opens up additional applications beside motion compensation. For instance, utilization for motion model establishment should be considered. In general, reconstructed image data is less limited in scope than raw projection data." I do not understand what this means.)
>
> Yes - in theory, for each reconstructed image volume there exists corresponding raw projection data.
> In practice, it is much harder for us to get the raw projection data, since this requires the interruption of the clinical routine. With motion model establishment we meant, that the motion estimation method can be used to identify predominant motion directions of the vessels at different heart phases over multiple image volumes (like a motion atlas). But we see that this is confusing and removed it from the paper.
>
> 4.	Regarding the second stated advantage: CNNs are also computationally expensive and require GPUs to train and apply them. If the authors wish to make an argument of computational efficiency, they should compare the running times, at least theoretical ones.
>
> You are right, there is merely an advantage with respect to computational effort for the image reconstruction as our approach only requires one back projection step. So far, total running times are not comparable (see also response 1). This is clarified in the paper, now.

---

### Review · AnonReviewer3 · 2018-05-09
**Review of Motion Estimation in Coronary CT Angiography Images using Convolutional Neural Networks**

**Rating:** 3
**Confidence:** 3

**Review:**

The authors investigated the feasibility of single-phase, image-based motion estimation by convolutional neural networks (CNNs). Based on simulated motion to artifact-free CT images, regression networks are trained to estimate underlying 2D motion vectors from axial coronary cross-sections. CNNs predict the motion direction and the motion strength with average
accuracies of 1.08◦ and 0.06 mm, respectively. This study looks very challenge and very important and experimental results are relatively extensive.

1. Why don't you compare mortion vector map of deformable registration with your CNN's estimation on voxel-by-voxel and evaluate the accuracy
2. What's the reason of angle and strength paritioning? Is there any possibility to make another overfitting?
3. Considering 80x80 patch size, is it reasonable to assume that the there is only one motion direction in patch? Did you only concern patch's center motion?
4. How do you evaluate the gold standard of motion in clinical study?
5. Is is good enough to estimate motion in 2D patch in axial slices?

**Special Issue:**

Yes

---

> ### Comment · ~Tanja_Elss1 · 2018-06-11
> **Re: Review of Motion Estimation in Coronary CT Angiography Images using Convolutional Neural Networks**
>
> Thank you for valuable remarks and for pointing out the following issues:
>
> 1.	Why don't you compare motion vector map of deformable registration with your CNN's estimation on voxel-by-voxel and evaluate the accuracy
>
> First, we have to extend our model to full 3D motion trajectories to allow for comparison with registration-based approaches. Then, this is indeed a great idea.
>
> 2.	What's the reason of angle and strength partitioning? Is there any possibility to make another overfitting?
>
> Our motivation for angle and strength partitioning in the phantom study is to test the interpolation capabilities of our networks with regard to unseen target motion directions and strengths.
> For the clinical study, we perform case-wise data separation to prevent for simple memorization of background intensities in patches of neighboring centerline points.
>
> 3.	Considering 80x80 patch size, is it reasonable to assume that the there is only one motion direction in patch? Did you only concern patch's center motion?
>
> Yes, the predicted motion vector relates to the patch center. The motion compensation experiment shows that it is not reasonable to assume that there is only one predominant motion direction in a 60x60 patch (see case 1 and 4 of Figure 7). But the fully convolutional design of our network also allows one to calculate motion vector maps.
>
> 4.	How do you evaluate the gold standard of motion in clinical study?
>
> Sorry, I am not entirely sure what you mean. Can you please reformulate your question?
>
> 5.	Is it good enough to estimate motion in 2D patch in axial slices?
>
> In this feasibility study we focused on 2D axial motion for simplicity.
> The proposed forward model is, in principle, extendable to arbitrary non-linear 3D motion trajectories. It needs to be reviewed to which extent the information content of the reconstructed image volumes is a limiting factor for estimation of more complex trajectories. Extension of our model to motion along the z-axis and turning motion will be the next steps.

---

### Review · AnonReviewer1 · 2018-05-10
**Well written interesting paper. Visual results look promising**

**Rating:** 4
**Confidence:** 1

**Review:**

This paper presents methods to use CNNs to predict the motion directions from 2D CT angiography patches for motion compensation. The problem is formulated into a multi-value regression problem with 2D image patch input. The motion blurred data used for training are synthesized by the MC-FBP model. The methods were tested with six cases from an in-house dataset. Both qualitative and quantitative results were reported.

Pros:
The paper is well written and easy to follow.
The proposed methods are interesting and sound.  The results look promising.

Cons:
The experiments fall short of the comparison with the previous baseline methods. It is not clear how the paper compares to the previous literature.
The paper only considers motions in the 2D plane. It is not described why 3D motions are not considered for motion compensation, especially for curvilinear structures.

**Special Issue:**

Yes

---

> ### Comment · ~Tanja_Elss1 · 2018-06-11
> **Re: Well written interesting paper. Visual results look promising**
>
> Thank you for your helpful comments.  We totally agree with your mentioned cons:
>
> 1.	The experiments fall short of the comparison with the previous baseline methods. It is not clear how the paper compares to the previous literature.
>
> Yes, the comparison with previous baseline methods will be a part of future research.
> In this paper, the feasibility of image-based motion estimation is investigated.
> The proposed motion compensation experiment is a proof of concept and demonstrates that the trained network also achieves reasonable results on data with real (non-simulated) motion.
> But, total running times and performances are not yet comparable.
> Further research has to be done to establish a full 3D motion compensation method.
>
> 2.	The paper only considers motions in the 2D plane. It is not described why 3D motions are not considered for motion compensation, especially for curvilinear structures.
>
> The proposed forward model is, in principle, extendable to arbitrary non-linear 3D motion trajectories. It needs to be reviewed to which extent the information content of the reconstructed image volumes is a limiting factor for estimation of more complex trajectories. Extension of our model to motion along the z-axis and turning motion will be the next steps.

---

### Comment · ~Bram_van_Ginneken1 · 2018-05-18
**Selection for longlist for special issue Medical Image Analysis**

Dear authors,

Congratulations on your acceptance to MIDL! We have selected your paper on the longlist for the Medical Image Analysis Special Issue. Please read this page:
https://midl.amsterdam/special-issue-in-medical-image-analysis/
Please answer the three questions that are listed on that page about your interest in submitting to the special issue, potential overlap with other publications, and related publications.

You can post your answer here directly below on openreview.net, or mail me directly at bram.vanginneken@radboudumc.nl.

Best regards, Bram

---

### Decision · Program_Chairs · 2018-05-15
**Paper17 Acceptance Decision**

Poster